# Effect of Microstructure on Coalescence-Induced Droplet Jumping Behavior of a Superhydrophobic Surface and Its Application for Marine Atmospheric Corrosion Protection



Zhengshen Chen [1,2,†], Xiaotong Chen [1,†], Yihan Sun [2,3], Guoqing Wang [1,*] and Peng Wang [2,3,*]

1 State Key Laboratory of Marine Resource Utilization in South China Sea, School of Materials Science and Engineering, Hainan University, Haikou 570228, China; 21220856000009@hainanu.edu.cn (Z.C.); chenxiaotong161@163.com (X.C.)
2 Key Laboratory of Marine Environmental Corrosion and Biofouling, Institute of Oceanology, Chinese Academy of Sciences, Qingdao 266071, China; sunyihan@qdio.ac.cn
3 Center for Ocean Mega-Science, Chinese Academy of Sciences, Qingdao 266071, China
* Correspondence: wangguoqing@hainanu.edu.cn (G.W.); wangpeng@qdio.ac.cn (P.W.)
† These authors contributed equally to this work.

**Abstract:** Coalescence-induced droplet jumping behavior (CIDJB) refers to the spontaneous jumping of droplets on a specific superhydrophobic surface (SS) without any external energy, which offers a new approach to the field of marine atmospheric corrosion protection by isolating corrosive media. In this study, a flower-like micro–nanocomposite structure SS (F-SS) and a sheet-like nanostructure SS (S-SS) were prepared on copper sheets by ammonia immersion and chemical vapor deposition. Firstly, we observed the microstructure characteristics of the samples and secondly analyzed its chemical composition and wettability. Moreover, the CIDJB was studied by simulated condensation experiments, and the influence of the microstructure on CIDJB was revealed. Meanwhile, the atmospheric corrosion resistance of samples was analyzed by electrochemical impedance spectroscopy (EIS) measurements, and the protection mechanism of SS through CIDJB was proposed. The results showed that the S-SS had a smaller solid–liquid contact area and lower interfacial adhesion, which is more conducive to CIDJB. Since a larger solid–liquid contact area requires greater interface adhesion energy for the droplets to overcome, droplet jumping behavior was not observed on the F-SS. Compared with the F-SS, the S-SS exhibited outstanding corrosion resistance due to the wettability transition of droplets by CIDJB, which facilitated the restoration of the air film to insulate the corrosive medium. The present study provides a reference for a marine atmospheric corrosion resistance technique through CIDJB on an SS.

**Keywords:** superhydrophobic; surface; atmosphere; corrosion; droplet; structure; area; energy

## 1. Introduction

Marine atmospheric corrosion is essentially electrochemical corrosion occurring between droplets or water film and a metal surface [1]. According to the bionics principle of the "lotus effect", a superhydrophobic surface (SS) is conducive to preventing the formation of water film on the material surface, which is mainly manifested as the deliquesced salt particles readily slide off the inclined SS under external forces [2]. Based on this anti-corrosion mechanism, more and more methods are being used to generate special hierarchical micro/nanostructures on metal substrates. Bai [3] used ferric chloride to etch a heat-transferred copper sheet and obtained a column-like structure SS after immersion in a stearic acid solution. Zheng et al. [4] prepared a vertically oriented few-layer graphene SS on Cu substrate by plasma-enhanced chemical vapor deposition technique. Cao et al. [5] constructed $Cu(OH)_2$ nano grass in situ on the surface of copper by simple anodizing and then modified it with fluoroalkyl silane to obtain a nanoneedle-like SS. However, the "lotus

effect" of the SS prepared by these methods is often achieved by external forces, which limits its application in marine environments.

A characteristic droplet-jumping phenomenon was first observed on an SS in 2009 [6] when merged droplets released excess surface energy and spontaneously jumped off the surface without requiring any external energy. Coalescence-induced droplet jumping behavior (CIDJB) has been widely studied and applied due to its various advantages, with its main applications being anti-frosting [7,8], self-cleaning [9], condensation heat transfer enhancement [10,11], and energy collection [12,13]. Theoretically, droplet jumping induced by condensation can prevent droplets from forming water films and spontaneously remove corrosive media, which lays the foundation for marine atmospheric corrosion protection. Previously, research on coalescence-induced droplet jumping behavior focused on droplet parameters, such as equal size [14], distribution [15], and initial velocity [16], but the current research direction is not limited to the droplet. Through nonequilibrium molecular dynamics simulation and energy-based theoretical analysis, Gao et al. [17] found that condensed droplets have different morphologies and self-propelling transitions in nucleation, growth, and coalescence stages, which actuate droplet migration and jumping. Tang et al. [18] studied a V-shaped SS with a triangular prism, which increased the jump speed and energy conversion efficiency by 80% and 210%, respectively, due to the reaction force exerted by the V-shaped sidewalls and the triprism. Huang et al. [19] studied the condensation and self-propelled jumping processes of droplets on the SS of a microcolumn structure with various dimensional parameters and found that the optimal spacings for width and the ratio of height to width for the microcolumn used for droplet jumping were about 0.6 and 1.0, respectively. By using a facile hydrothermal method to fabricate a nanorod-like structure superhydrophobic surface on a zinc substrate, and then modifying the superhydrophobic surface with three different low-surface-energy materials (1H,1H,2H,2H-perfluorodecyltriethoxysilane (PFDS), decyltriethoxysilane (DTS), and stearic acid (SA)), Liu et al. [20] obtained a superhydrophobic surface with different surface energies and analyzed the relevant mechanisms from the perspective of energy, revealing the key factors that affect surface energies and affect the jumping behavior of droplets. Only a few pieces of research have focused on the structure of the superhydrophobic surface itself, especially the scale of the composite structure [21–23].

In this study, a flower-like micro–nanocomposite structure SS (F-SS) and a sheet-like nanostructure SS (S-SS) were constructed on copper substrates by ammonia immersion and chemical vapor deposition. Based on the reasonable preparation of two kinds of superhydrophobic surfaces that are beneficial and not beneficial to coalescence-induced droplet jumping behavior, a new mechanism of atmospheric corrosion protection was revealed by studying the correlation of the surface structure, droplet jumping behavior, and atmospheric corrosion resistance of the two surfaces. It was found that, compared with F-SS, the S-SS had a lower interfacial adhesion energy ($E_w$) due to the smaller solid–liquid contact area, which is conducive to droplet jumping behavior. The SS with droplet jumping behavior presented a superior anti-corrosion performance due to the droplet-jumping-induced wetting transition, which promoted the recovery of the insulation and barrier character of the air film. The present study provides a reference for a marine atmospheric corrosion resistance technique through CIDJB on an SS.

## 2. Experimental Section

### 2.1. Materials

1H,1H,2H,2H-perfluorodecyltriethoxysilane (PFDS, 97%) was purchased from Sigma-Aldrich, St. Louis, MO, USA. The Cu foil (99.5 wt.%) was purchased from Guangdong Fuye Copper Industry Co., Ltd, Guangdong, China. Ammonium hydroxide ($NH_3 \cdot H_2O$), hydrochloric acid (HCl), ethanol ($C_2H_6O$), and acetone ($C_3H_6O$) were purchased from Macklin Reagent Co., Ltd, Shanghai, China.

## 2.2. Preparation of Superhydrophobic Surfaces

The F-SS and the S-SS were constructed on copper substrates by ammonia immersion and chemical vapor deposition, respectively, as shown in Figure 1. Specifically, the copper foil was ground with sandpaper to 2000 grit, then ultrasonically washed with acetone and anhydrous ethanol in turn to remove the grease, and then ultrasonically cleaned with 2.0 M HCl to remove the surface oxides. The washed copper foil was then dipped into 0.03 M $NH_3H_2O$ solution, soaked at 50 °C for 48 h to prepare the F-SS, and soaked at 50 °C for 72 h to prepare the S-SS, respectively. After the reaction was complete, the samples were taken out and rinsed with water and anhydrous ethanol in turn. Finally, the sample was placed in an autoclave; 15 μL PFDS was added and reacted at 120 °C for 2 h. After that, the sample was retrieved from the autoclave and heated at 150 °C for 1 h.

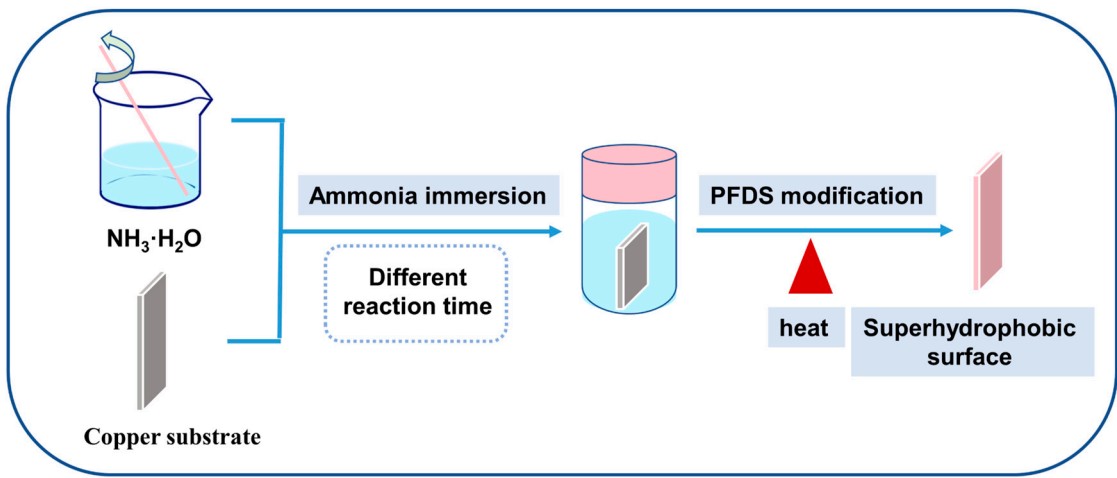

**Figure 1.** The schematic of the prepared processes of the superhydrophobic surfaces.

## 2.3. Characterization

The microstructure of the superhydrophobic surfaces was observed using a high-resolution field emission scanning electron microscope (FE-SEM, Hitachi S4800, Hitachi, Tokyo, Japan). A secondary electron detector was used in the FE-SEM analysis.

A contact-angle-measuring instrument (CA, Powereach JC2000C1, Shanghai Zhongchen Digital Technology Equipment Co., Ltd, Shanghai, China) was used to measure the contact angle of the surface.

The surface crystal structure analysis was recorded by an X-ray diffractometer (XRD, Rigaku Ultima IV, Japan Science company, Tokyo, Japan) with Cu Kα radiation (λ = 1.54 Å).

The chemical composition of the superhydrophobic surfaces was analyzed by X-ray electron spectroscopy (XPS, Escalab 250Xi, Thermo Corporation, Waltham, MA, USA), and Al Kα rays were tested as excitation sources.

## 2.4. Simulated Condensation Experiments

The jumping behavior of condensate droplets on the SS was recorded by a digital optical microscope (HiROX RH 2000, HIROX Corporation, Tokyo, Japan). The specific process was as follows: The copper block was placed in crushed ice for cooling, and then the sample was placed horizontally on its surface (about 1 °C) with a condensing environment temperature of 21 ± 2 °C and relative humidity of 65 ± 5%. In the meantime, the appropriate focal length was adjusted to take an image of the condensation droplet jumping behavior at 1 s intervals for 999 consecutive shots.

### 2.5. Electrochemical Measurements

The Gamry Electrochemical Workstation (Reference 3000) was used to perform electrochemical impedance spectroscopy (EIS) measurements. The measurements were conducted in a three-electrode cell with the platinum sheet as a counter electrode, an Ag/AgCl (3 M KCl) electrode as a reference electrode, and the SS as a working electrode. The experiments were performed in a 3.5 wt.% NaCl solution. The condensation conditions on the superhydrophobic surface in the electrochemical experiment were consistent with those in the simulated condensation experiment. The electrochemical test area was $1 \times 1$ cm$^2$, and the frequency range was set at $10^{-2}$~$10^5$ Hz with an excitation AC voltage of 10 mV. The test was conducted three times for each condition to ensure the repeatability of the results.

## 3. Result and Discussion

### 3.1. Morphology, Composition, and Wettability

The CIDJB on an SS is related to their microstructures. The morphological images of the SS are shown in Figure 2. The SS that was fabricated in 0.03 M NH$_3$H$_2$O solution at 50 °C for 48 h presented a flower-like structure (F-SS). It was a micro–nanocomposite structure. The bottom layer was densely packed with nanosheets about 60–100 nm in diameter, and the top layer was randomly distributed with many "flowers" about 1–3.5 μm in diameter formed by tightly stacked nanosheets. In contrast, the superhydrophobic surface reacted for 72 h, resulting in a sheet-like structure (S-SS). It was a nanostructure with evenly distributed nanosheets about 90–150 nm in diameter. The FE-SEM images of the samples soaked at 50 °C for different times are shown in Figure S1. We speculated that the copper oxide formed on the surface of the copper sheet existed in the F-SS during the early stage of soaking in ammonia. As the reaction progressed, oxygen and ammonia were consumed in the bottle, and copper oxides began to form in the S-SS. In the later period, as the nanosheet grew, the "flowers" on the top layer began to gradually fall off, thus transforming the F-SS into an S-SS.

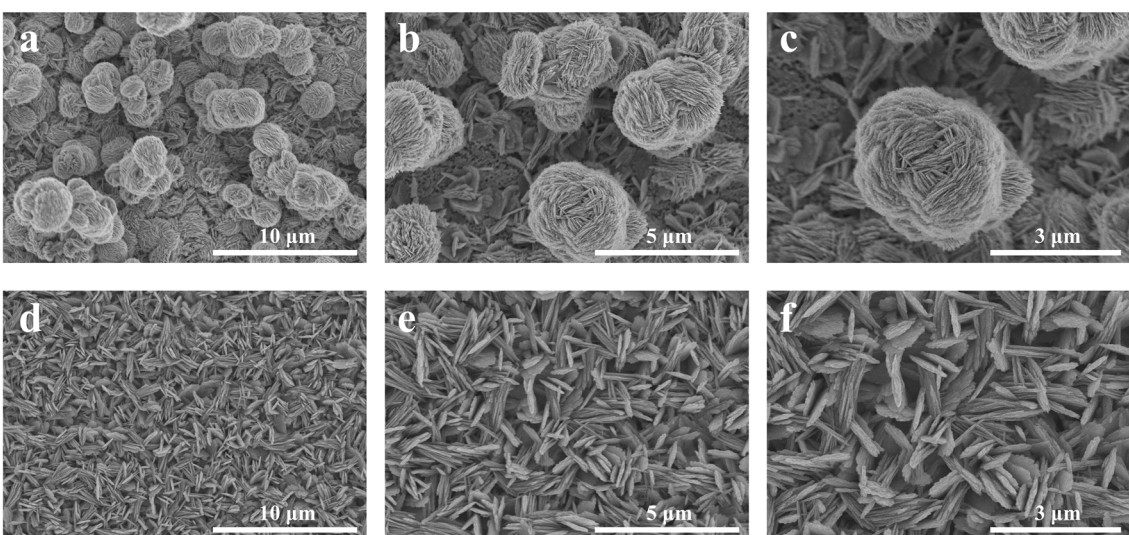

**Figure 2.** FE-SEM morphology images of the (**a**–**c**) F-SS and (**d**–**f**) S-SS.

As shown in Figure 3, the XRD results indicated that CuO and Cu$_2$O were the main compositions of the SS. The diffraction peaks of samples at 35.5° and 38.6° could be attributed to the [002] and [111] planes of the CuO substance (JCPD S no. 48-1548), respectively. The diffraction peaks of the two samples located at 36.7° could be attributed to the [111] plane of Cu$_2$O (JCPDS no. 05-0667). The diffraction peaks for the remaining markers were all derived from the Cu substrate (JCPDS no. 04-0836). In general, oxygen in the air reacts with ammonia and copper to form a Cu–ammonia complex, which improves the reducibility of copper and accelerates the oxidation of copper [24]. Cu$_2$O was formed on

the surface of the copper first and then oxidized to CuO. The chemical reaction involved was as follows [25–27]:

$$2\,Cu + 8\,NH_3\cdot H_2O + O_2 \longrightarrow 2\,[Cu(NH_3)_4](OH)_2 + 6\,H_2O \tag{1}$$

$$2\,Cu + 2\,OH^- \longrightarrow Cu_2O + H_2O + 2\,e^- \tag{2}$$

$$Cu_2O + 2\,OH^- \longrightarrow 2\,CuO + H_2O + 2\,e^- \tag{3}$$

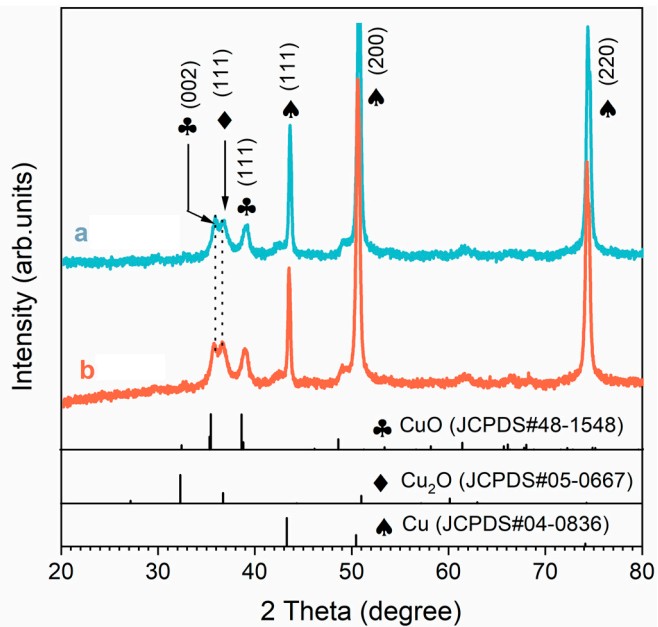

**Figure 3.** XRD patterns of the blue line (a) F-SS and the red line (b) S-SS.

XPS results indicated that the PFDS successfully formed a self-assembled monolayer on the surface of both samples, as shown in Figure 4 and Table 1. The XPS results showed that the two samples were composed of elemental Cu, O, C, F, and Si. For both the samples, the C 1s spectrum was resolved into four peaks with binding energies of 293.6, 291.4, 285.3, and 284.6 eV, which were consistent with the -CF$_3$, -CF$_2$, -C–CF$_x$, and -C–C bonds, respectively [27]. The -C–F bond at 688.7 eV could be recognized from the F 1 s spectra [28]. The binding energy at 102.1 eV was consistent with the -Si–O bond on the F-SS and the S-SS [29]. In addition, the atomic ratio of elements on the F-SS and the S-SS could be calculated as C/Cu/F/O/Si = 23/11.26/3.96/43.39/18.39 and C/Cu/F/O/Si = 22.85/12.2/4.02/41.31/19.62, respectively. The results above showed that the non-polar molecule -CF$_3$ and -CF$_2$ composed of PFDS successfully grafted to the SS during modification. The modification principle is as follows [30–32]: The functional groups of triethoxysilane present in PFDS engage in a chemical reaction with hydroxyl groups on the copper substrate, leading to the establishment of a covalent bond. Consequently, the perfluorinated carbon chains within PFDS reposition themselves in a manner that directs them away from the substrate surface, generating a self-assembled monolayer. The presence of carbon–fluorine bonds induces a significant reduction in the surface energy, thereby conferring superhydrophobicity upon the surface [33].

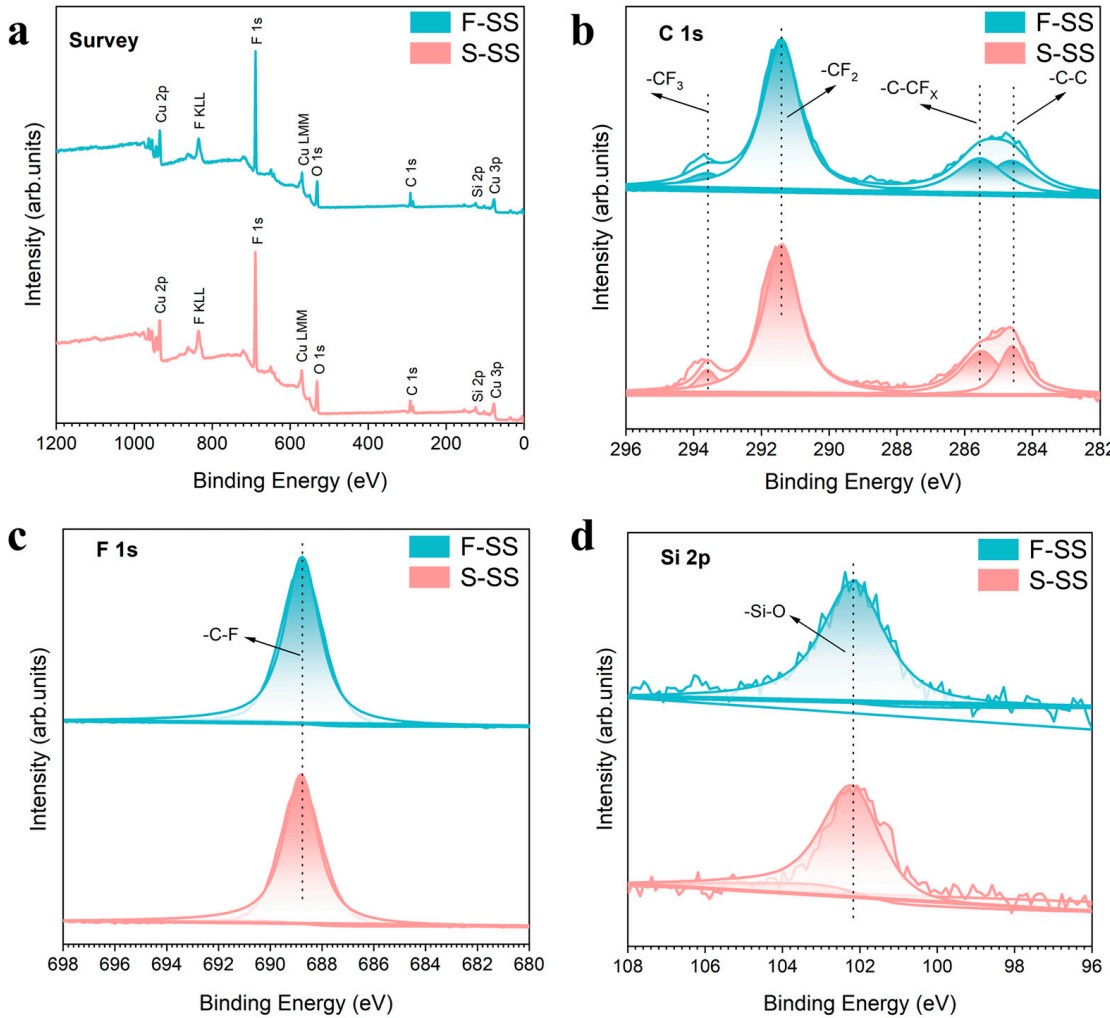

**Figure 4.** XPS spectra of the F-SS and S-SS: (**a**) survey spectra, (**b**) C 1s spectra, (**c**) F 1s spectra, and (**d**) Si 2p spectra.

**Table 1.** The peak binding energies values and FWHM fit parameters of the F-SS and the S-SS.

|  | Samples | C 1s | | | | F 1s | Si 2p |
|---|---|---|---|---|---|---|---|
|  |  | -CF$_3$ | -CF$_2$ | -C-CF$_x$ | -C-C | -C-F | -Si-O |
| F-SS | Binding Energy (eV) | 293.6 | 291.4 | 285.3 | 284.6 | 688.7 | 102.1 |
|  | FWHM | 1.31 | 1.29 | 1.73 | 1.70 | 1.61 | 1.77 |
| S-SS | Binding Energy (eV) | 293.6 | 291.4 | 285.3 | 284.6 | 688.7 | 102.1 |
|  | FWHM | 1.31 | 1.27 | 1.75 | 1.71 | 1.55 | 1.75 |

Fowkes theory was used to calculate the surface energy of different samples [34,35]:

$$\gamma_{sv} = \frac{\gamma_{lv}^2 (1 + cos\theta_Y)^2}{\gamma_{lv}^d} \tag{4}$$

The $\gamma$ represents surface energy [36], with subscripts s, l, and v corresponding to the solid, liquid, and gas phases, respectively.

Taking water as an example, substitute the value of $\gamma_{lv}$ and $\gamma_{lv}^{d}$ [34] for water at 20 °C and $\theta_Y = 119°$ [37] into Equation (4) to obtain $\gamma_{svFS} = \gamma_{svSS} = 64.44$ mJ/m$^2$. Superhydrophobic surfaces with different microstructures have the same chemical composition and surface energy [27].

The contact angle measurement results of the SS are shown in Figure S2. The contact angles of the F-SS and the S-SS after ammonia immersion were close to 0°. After soaking in ammonia, an intrinsic hydrophilic copper oxide with a rough structure was formed on the surface of the sample, resulting in surface superhydrophilicity. After chemical vapor deposition, the wettability of the surface was changed due to the combination of the surface roughness increasing and the surface energy decreasing [2,20]. The F-SS and the S-SS displayed superhydrophobicity, with respective contact angles of $156 \pm 2°$ and $160 \pm 2°$. According to the Cassie–Baxter equation [38], the solid-liquid area ratio ($f_1$) and the vapor-liquid area ratio ($f_2$) can be obtained from the equation:

$$cos\theta_c = f_1 cos\theta - f_2 \tag{5}$$

where $\theta$ and $\theta_c$ are the intrinsic contact angle and apparent contact angle, respectively, and $f_1 + f_2 = 1$.

The intrinsic contact angle for solid surfaces with a -CF$_3$ bond is usually 119° [37], and the $f_2$ of the F-SS and the S-SS was 0.8322 and 0.8830, respectively. It could be seen that a certain air layer could be captured in the gaps between the nanosheets on the SS, and the existence of the air film was an important reason for the superhydrophobicity of the surface [27,39]. The $f_1$ of the F-SS and the S-SS was 0.1678 and 0.1170, respectively, with the former exhibiting a higher $f_1$.

### 3.2. Coalescence-Induced Droplet Jumping Behavior

We observed the surface morphology of the samples after the simulated condensation experiments, and the FE-SEM images are shown in Figure S3. Compared with FE-SEM images before condensation, no significant changes were found. The F-SS, the S-SS, and the bare substrate (BS) had different droplet jumping behaviors in the simulated condensation experiment, as shown in Figure 5. It could be seen that the two droplets on the S-SS jumped at the instant of merging, indicating the realization of CIDJB. In contrast, droplets on the F-SS and the bare substrate did not disappear after merging but formed larger-diameter droplets or larger areas of the water film. A video related to these two different droplet behaviors was recorded and is available in Supporting Information S4. As shown in Figure S3, the surface morphology of the sample did not change significantly after the simulated condensation experiment. The change of wetting coverage (the sum of all observed droplet areas/the microscope observed field of view) and the condensation suppression efficiency (the ratio of the number of drops that jump after merging to the total number of droplets that merge) are shown in Figure 6. The wetting coverage in the BS continued to increase, and the condensation suppression efficiency was zero. The two samples had similar droplet surface coverage at the initial stage of condensation. For the F-SS, the wetting coverage of droplets increased with time, and the condensation suppression efficiency was always zero. However, for the S-SS, the wetting coverage of droplets was maintained at about 10–20%, and condensation suppression efficiency was always greater than 80%. Concerning the S-SS, the droplets that grew were able to jump from the surface after merging, which led to the wetting coverage being controlled at a relatively stable low level. However, for the F-SS, the droplets continued to grow into a larger-diameter droplet after merging and finally formed a large area of water film similar to the bare substrate. The above results showed that the CIDJB of the S-SS could significantly prevent droplets from forming water films.

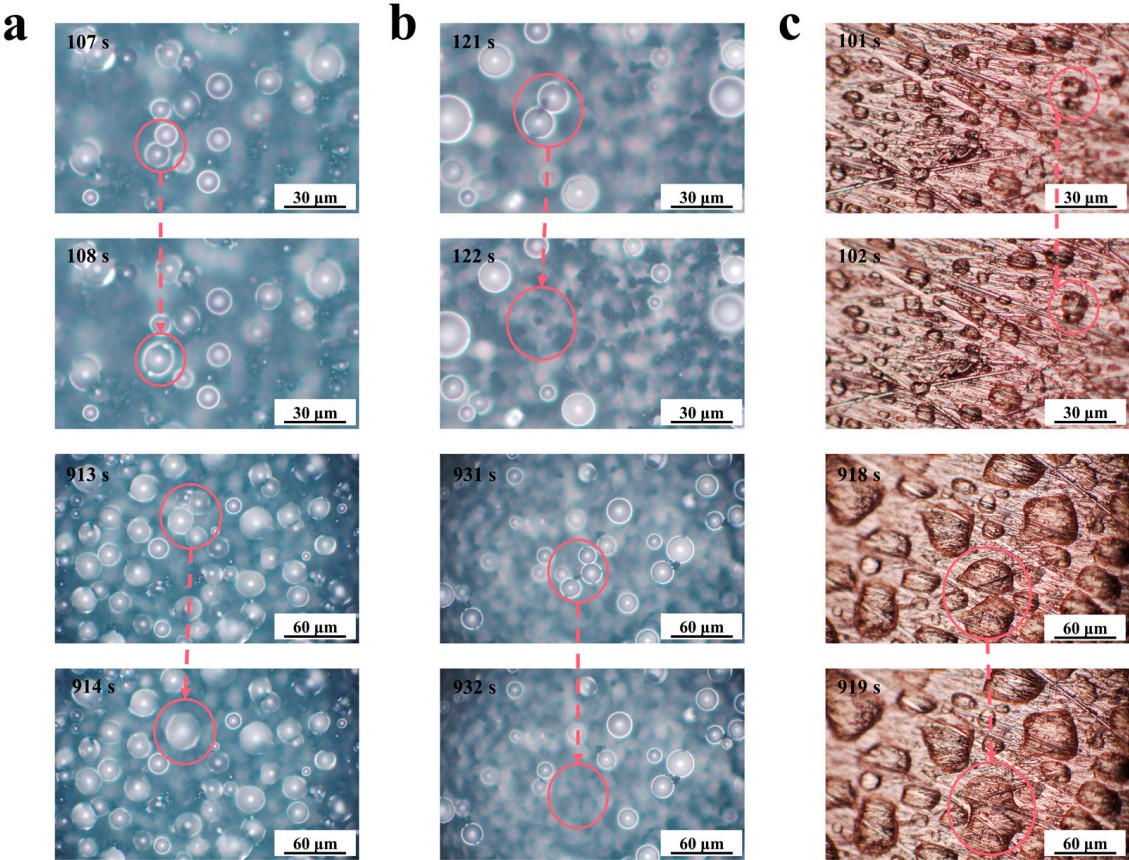

**Figure 5.** Optical top-view images of condensate droplets at different times on the (**a**) F-SS, (**b**) S-SS, and (**c**) BS. The red circles highlight the droplets before and after the merging.

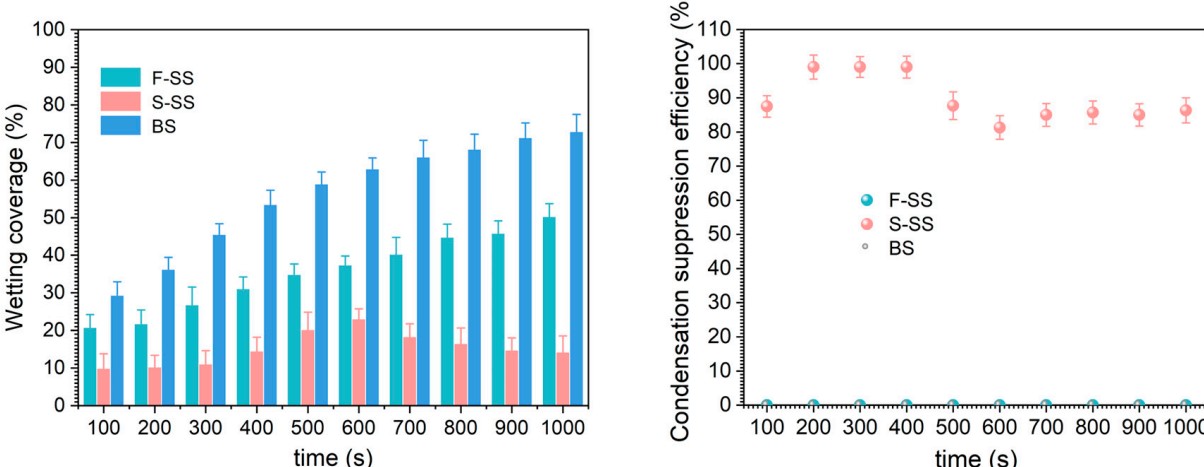

**Figure 6.** The wetting coverage (**a**) and the condensation suppression efficiency (**b**) of the F-SS, S-SS, and BS.

In general, the jumping energy equation of droplet coalescence is determined by the kinetic energy ($E_k$), satisfying the equation [6,40,41]:

$$E_k = \Delta E_s - E_w - E_{vis} - E_h \qquad (6)$$

The surface energy ($\Delta E_\text{s}$) released by the droplet merger can be calculated as [42]:

$$\Delta E_\text{s} = 2\pi\gamma_\text{lv} R^2 \left(2 - 2cos\theta - C(f)sin^2\theta\right) - 4\pi\gamma_\text{lv}\left(2R^3\right)^{\frac{2}{3}} \tag{7}$$

The viscous dissipated energy ($E_\text{vis}$) per droplet can be calculated as [43]:

$$E_\text{vis} = 36\pi\mu\left(\gamma_\text{lv} R^3 \rho^{-1}\right)^{\frac{1}{2}} \tag{8}$$

The gravitational potential energy ($E_\text{h}$) of the droplet can be calculated as [44]:

$$E_\text{h} = mg\left[\left(\frac{3V}{4\pi}\right)^{\frac{1}{3}} - \frac{r(3 + cos\theta)(1 - cos\theta)}{4(2 + cos\theta)}\right] \tag{9}$$

For the droplet with a diameter larger than 500 nm, the $E_\text{vis}$ becomes relatively small compared to the $E_\text{k}$ and can be ignored [45]. Similarly, for the droplet with a diameter smaller than the capillary length (2.7 mm), the $E_\text{h}$ can be ignored [46]. Therefore, the key factor to whether a droplet can jump depends on whether the $\Delta E_\text{s}$ can overcome interfacial adhesion energy ($E_\text{w}$) and convert it into $E_\text{k}$.

The $E_\text{w}$ of the droplet can be calculated as [33,47]:

$$E_\text{w} = \gamma_\text{lv}(1 + cos\theta_\text{Y})A_\text{sl} \tag{10}$$

The $A_\text{sl}$ represents the influence of the solid–liquid contact area on interfacial adhesion before droplet merging [27,33]. Both samples had the same chemical composition, and the interfacial adhesion was directly affected by the solid–liquid contact area. The F-SS had a layer of 1–3.5 µm "flowers" made of stacked nanosheets. The gaps between the random and tightly packed nanosheets could be regarded as nanoscale channels and vacancies [2,48], which provided a high capillary effect for small droplets. The capillary effect increased the contact area, which led to greater interfacial adhesion. Without the restriction of a larger top layer, the S-SS had a smaller solid–liquid contact area and $E_\text{w}$ compared with the F-SS, which was beneficial to the realization of CIDJB.

### 3.3. Marine Atmospheric Corrosion Protection Performance

First of all, since our main application is a marine atmospheric corrosion environment, a 3.5% NaCl solution could be used to simulate the marine environment. Secondly, since it is difficult to conduct electrochemical experiments directly in an open atmosphere, the atmospheric corrosion resistance of the SS was generally detected in a 3.5% NaCl solution. The electrochemical experiments of superhydrophobic surfaces tested in solution truly reflect their barrier effect in an open atmosphere because water droplets in solution and atmospheric environments have the same contact mode on SS. The choice of EIS test frequency was based on the conditions of the instrument, and other researchers also used this frequency for testing [49–51]. The EIS measurements of the F-SS, the S-SS, the bare substrate (BS), and the DS (degassed SS through ethanol immersion) are shown in Figure 7. In general, the impedance magnitude ($|Z|$) can be used as an evaluation of SS atmospheric corrosion protective performance of half quantitative indicators, and a larger $|Z|$ value at a frequency of 0.01 Hz represents excellent corrosion resistance [52,53]. Before the condensation, the $|Z|$ values of the F-SS and the S-SS were about one thousand times larger than that of the BS, confirming that the two kinds of structure of SS had outstanding corrosion protection performance. At the end of condensation, the $|Z|$ values of the F-SS and S-SS showed disparate degrees of decline. Among them, the $|Z|$ value of the F-SS fell sharply to $10^4$ Ω cm$^2$. The $|Z|$ value of the S-SS could remain at $10^7$ Ωcm$^2$. The above results showed that the S-SS had a more stable corrosion resistance than the F-SS after the condensation. The $|Z|$ values of DS were significantly less than that of the SS, indicating that the air layer on the SS played a vital role in the isolation of corrosive media.

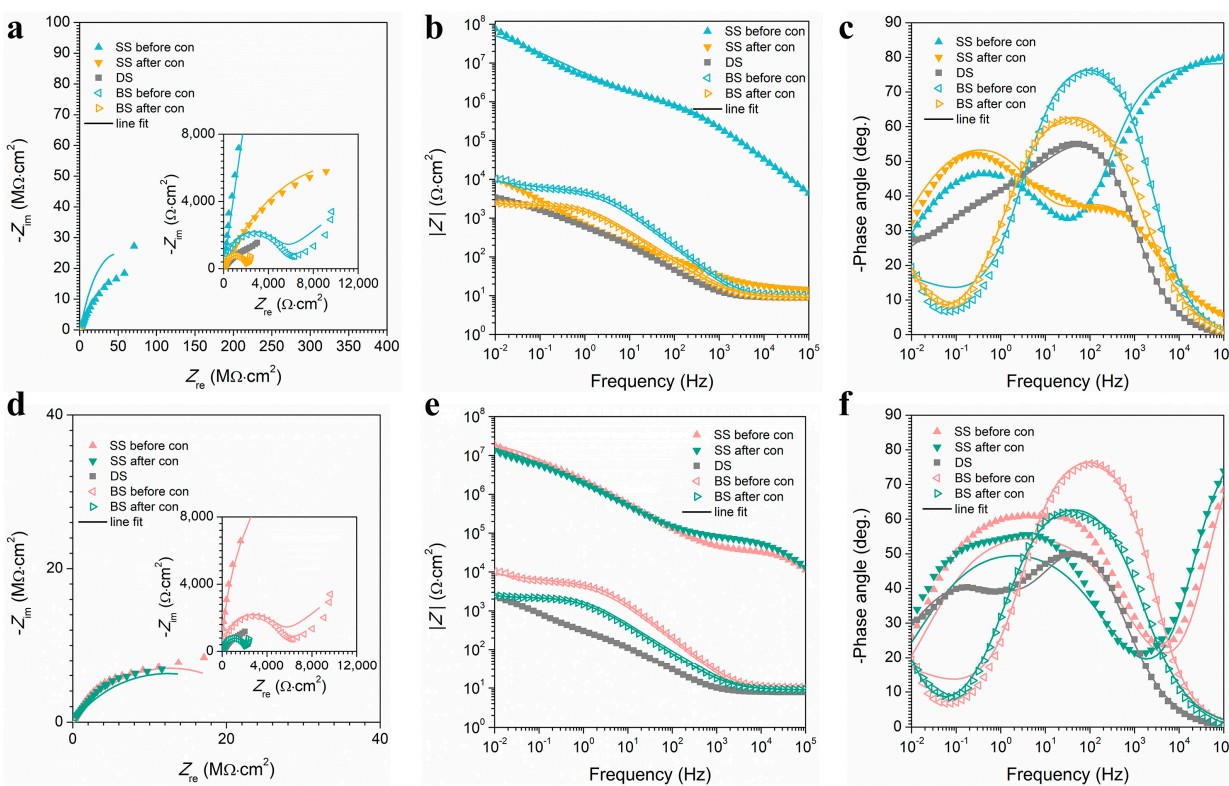

**Figure 7.** EIS results for (**a**–**c**) F-SS and (**d**–**f**) S-SS. (**a**,**d**) The Nyquist curve, (**b**,**e**) the Bode |Z| versus frequency curve, and (**c**,**f**) the Bode phase angle versus frequency curve.

We used different equivalent circuit diagrams to fit the EIS results, as shown in Figure S4. The data obtained by fitting are shown in Table 2. The film capacitance ($C_f$) in the table could be calculated as [39,48]:

$$C_f = R_s^{\frac{1-n}{n}} Y_f^{\frac{1}{n}} \tag{11}$$

After simulated condensation experiments, the $C_f$ of the F-SS and the S-SS increased, while the $R_f$ decreased. It indicates that droplets could penetrate the microstructure of the surface during the simulated condensation experiments.

The double-layer capacity ($C_{dl}$) and the $R_{ct}$ are indexes to evaluate the corrosion resistance of metals. In general, a low value of $C_{dl}$ indicates a smaller number of available sites for electrochemical reactions, thus reducing the corrosion rate, while a high value of $R_{ct}$ indicates good electrocatalytic activity and low electron transfer rates, which help protect metal surfaces from corrosion. The $C_{dl}$ and corrosion inhibition efficiency ($\eta$) are calculated as [54,55]:

$$C_{dl} = (Y_{dl})^{\frac{1}{n}} \left( \frac{1}{R_s} + \frac{1}{R_{ct}} \right)^{\frac{n-1}{n}} \tag{12}$$

$$\eta(\%) = \frac{R_{ct} - R_{ct}^0}{R_{ct}} \times 100 \tag{13}$$

**Table 2.** The fitted electrochemical parameters of the EIS results.

| Samples | $R_{\mathrm{s}}$ (Ω cm²) | $Q_{\mathrm{f}}$ | | $R_{\mathrm{f}}$ (Ω cm²) | $Q_{\mathrm{dl}}$ | | $R_{\mathrm{ct}}$ (Ω cm²) | $W$ (Ω⁻¹ s⁰·⁵ cm⁻²) | $C_{\mathrm{f}}$ (μF cm⁻²) | $C_{dl}$ (μF cm⁻²) | $\eta$ (%) |
|---|---|---|---|---|---|---|---|---|---|---|---|
| | | $Y_{\mathrm{f}}$ (F cm⁻²) | $n_{\mathrm{f}}$ | | $Y_{\mathrm{dl}}$ (F cm⁻²) | $n_{\mathrm{dl}}$ | | | | | |
| F-SS before con | 13.75 ± 0.890 | (1.98 ± 0.871) × 10⁻⁹ | 0.8740 ± 0.014 | (9.586 ± 0.178) × 10⁵ | (6.393 ± 0.156) × 10⁻⁸ | 0.5979 ± 0.007 | (9.709 ± 0.134) × 10⁷ | - | 1.609 × 10⁻⁴ | 5.407 × 10⁻⁶ | 99.99 |
| F-SS after con | 13.30 ± 0.781 | (2.388 ± 0.776) × 10⁻⁴ | 0.5910 ± 0.009 | 307.6 ± 12.60 | (1.978 ± 0.009) × 10⁻⁴ | 0.7321 ± 0.013 | (2.262 ± 0.113) × 10⁴ | - | 4.459 | 22.49 | 83.36 |
| S-SS before con | 9.952 ± 0.786 | (1.664 ± 0.897) × 10⁻¹⁰ | 0.9812 ± 0.011 | (2.993 ± 0.134) × 10⁵ | (1.365 ± 0.131) × 10⁻⁷ | 0.6395 ± 0.009 | (2.523 ± 0.147) × 10⁷ | - | 1.130 × 10⁻⁴ | 6.727 × 10⁻⁵ | 99.98 |
| S-SS after con | 11.17 ± 0.889 | (2.820 ± 0.832) × 10⁻¹⁰ | 0.9868 ± 0.013 | (2.240 ± 0.119) × 10⁵ | (1.808 ± 0.123) × 10⁻⁷ | 0.6284 ± 0.011 | (2.456 ± 0.151) × 10⁷ | - | 2.170 × 10⁻⁴ | 7.756 × 10⁻⁵ | 99.98 |
| F-DS | 8.952 ± 0.789 | (6.452 ± 0.690) × 10⁻⁵ | 0.8699 ± 0.016 | 151.6 ± 9.36 | (6.275 ± 0.007) × 10⁻⁴ | 0.5643 ± 0.009 | 4757 ± 11.78 | - | 21.15 | 11.47 | - |
| S-DS | 7.699 ± 0.897 | (2.842 ± 0.756) × 10⁻⁴ | 0.7579 ± 0.015 | 200.8 ± 15.3 | (1.323 ± 0.006) × 10⁻³ | 0.5669 ± 0.008 | 4550 ± 16.31 | - | 40.17 | 39.73 | - |
| BS before con | 10.26 ± 0.981 | (1.432 ± 0.981) × 10⁻⁵ | 0.9089 ± 0.011 | 3615 ± 22.1 | (2.657 ± 0.011) × 10⁻⁴ | 0.6579 ± 0.011 | 5352 ± 18.11 | (2.274 ± 0.112) × 10⁻⁴ | 5.912 | 12.32 | - |
| BS after con | 8.871 ± 0.887 | (9.440 ± 0.753) × 10⁻⁵ | 0.7687 ± 0.010 | 2273 ± 21.9 | (1.775 ± 0.013) × 10⁻² | 0.5824 ± 0.007 | 3764 ± 10.09 | (1.880 ± 0.009) × 10⁻³ | 11.20 | 4708 | - |

In Equation (13), $R_{ct}^0$ and $R_{ct}$ corresponds to the charge-transfer resistance of the BS and SS, respectively. It is worth noting that the F-SS and the S-SS had outstanding corrosion resistance before condensation. Their $C_{dl}$ was $5.407 \times 10^{-6}$ and $6.727 \times 10^{-5}$ µF cm$^{-2}$, respectively, with a corresponding $\eta$ of 99.99% and 99.98%, respectively. After the simulated condensation experiments, the $C_{dl}$ of the S-SS increased slightly from $6.727 \times 10^{-5}$ to $7.756 \times 10^{-5}$ µF cm$^{-2}$, while the $R_{ct}$ and $\eta$ remained roughly the same. The $C_{dl}$ of the F-SS rose substantially from $5.407 \times 10^{-6}$ to 22.49 µF cm$^{-2}$, the $R_{ct}$ dropped dramatically from $9.709 \times 10^7$ to $2.262 \times 10^4$ Ω cm$^2$, and the $\eta$ also decreased from 99.99% to 83.36%. The $R_f$ of the F-SS decreased significantly after condensation, while the $R_f$ of the S-SS remained unchanged. The $C_{dl}$ of the DS was higher than that of the BS, indicating that the rougher surface was more conducive to the generation of active sites. Meanwhile, the $R_{ct}$ of the DS was lower than that of the BS, which was attributed to the fact that the penetration of water promotes the migration of corrosive substances such as Cl$^-$, thus further increasing the electrochemical reaction rate. These results are consistent with those of the $|Z|$ value semi-quantitative analysis. To sum up, it was confirmed that the S-SS had a superior corrosion resistance than the F-SS after condensation.

### 3.4. Atmospheric Corrosion Resistance Mechanisms

It has been demonstrated that a complete air film on the SS is a crucial prerequisite for achieving atmospheric corrosion protection [27]. The known principle of total reflection of light is one of the effective methods to evaluate the superhydrophobic surface in underwater air film [56,57]. When incident light strikes the SS, a clear reflection indicates the air layer is intact, while a blurry or absent reflection indicates a damaged or defective air layer. Since droplets contact the SS similarly in both solution and atmospheric environments [48], the samples were quickly put into solution after condensation to observe the total reflection of light and to analyze the influence of the condensation process on the superhydrophobic surface air layer, as shown in Figure 8. Before condensation, it could be seen that the air layer on the surface of both samples was intact and reflected all light. At the end of condensation, the S-SS remained bright, while the F-SS changed from bright to dull, indicating that the air layer on the F-SS was impaired during the condensation process. Therefore, we believe that the droplet jumping behavior was beneficial for the SS to keep the air film intact.

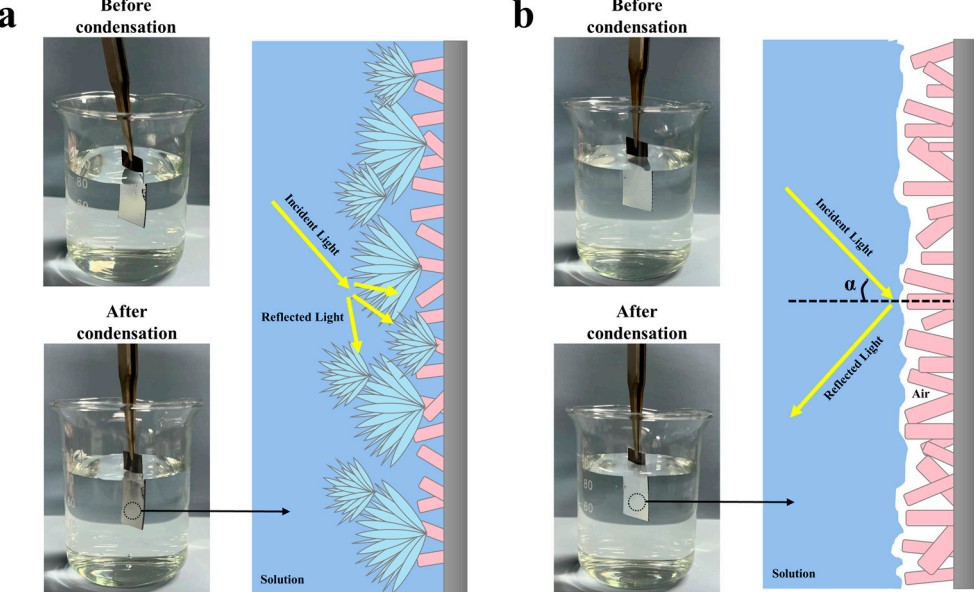

**Figure 8.** Digital photographs of the (**a**) F-SS and (**b**) S-SS in solution and schematic of light total reflection phenomenon.

At the beginning of condensation, the formed droplets contacted the surface in the Wenzel state or partially wet state, resulting in the gradual occupation of the air film. For the F-SS, droplets merged into larger droplets after contact, and the merged and grown droplets always existed in a Wenzel or partially wet state, as shown in Figure 9a. With the nucleation and growth of the droplets, the wetting coverage increased, leading to the disappearance of the air layer, which led to the occurrence of a corrosion reaction. For the S-SS, droplets jumped after the merger, and the jumping droplets could take away the trapped droplets in the nanosheet gap, thus facilitating the transition of the condensate droplets from the Wenzel or partially wetted state to the Cassie state [6], which facilitated the restoration of the air film, as shown in Figure 9b. The restoration of the air film insulates the corrosive medium and improves the atmospheric corrosion of the superhydrophobic surface.

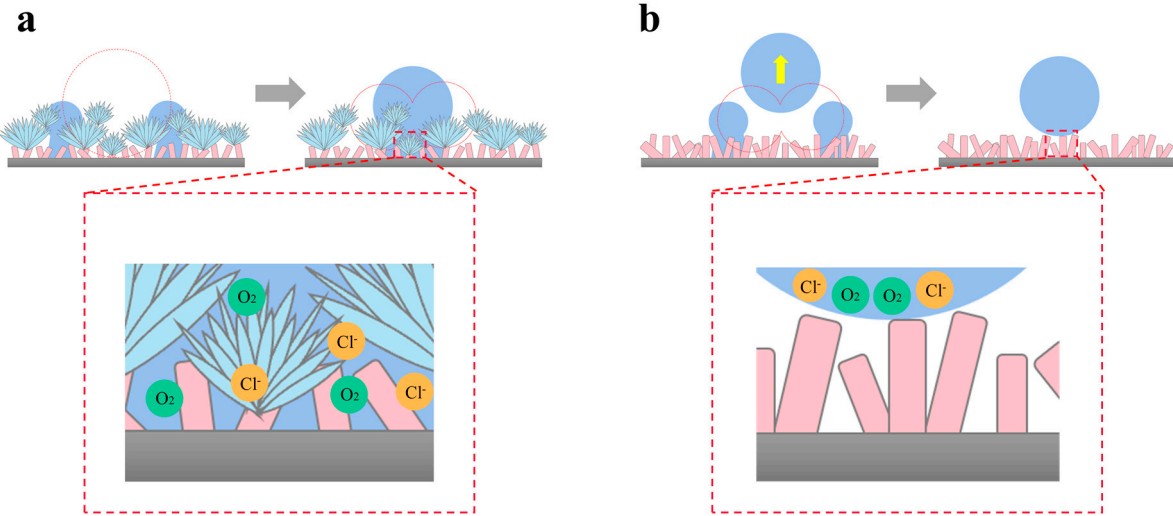

**Figure 9.** The anti-corrosion mechanism of superhydrophobic surfaces based on coalescence-induced droplet jumping behavior: (**a**) the F-SS and (**b**) the S-SS.

## 4. Conclusions

In this study, a flower-like micro–nanocomposite structure SS (F-SS) and a sheet-like nanostructure SS (S-SS) were constructed on copper substrates by ammonia immersion and chemical vapor deposition. The morphology, composition, and wettability of the samples were characterized. The differences in droplet jumping behavior of the two surfaces were investigated from the perspective of microstructure and energy. Meanwhile, the atmospheric corrosion resistance of samples was analyzed, and a protection mechanism of SS through CIDJB was proposed. The main conclusions are as follows:

(1) The contact angles of the two superhydrophobic surfaces were 156° and 160°, respectively;

(2) It was found that the S-SS could realize the CIDJB, the wetting coverage was maintained at about 10–20%, and the condensation suppression efficiency was greater than 80% in the simulated condensation experiments;

(3) The results of the EIS measurements showed that the corrosion inhibition efficiency ($\eta$) of the S-SS remained at 99.98% before and after condensation, while the $\eta$ of the F-SS decreased from 99.99% to 83.36%. These differences resulted from the different microstructures of the two superhydrophobic surfaces. Compared to the S-SS, the F-SS had a layer of 1–3.5 μm "flowers" composed of stacked nanosheets, and the larger solid–liquid contact area resulted in higher interfacial adhesion ($E_w$). The lower $E_w$ was required to be overcome; thus, it was beneficial for the S-SS to realize the droplet jumping behavior;

(4) The S-SS exhibited excellent corrosion resistance due to the wettability transition of droplet jumping behavior induced by coalescence, which facilitated the restoration of the air film.

**Supplementary Materials:** The following supporting information can be downloaded at: https://www.mdpi.com/article/10.3390/met13081413/s1, Figure S1: FE-SEM images of samples soaked in ammonia at 50°C for (a) 24 h, (b) 48 h, (c) 60 h, and (d) 72 h; Figure S2: The CA of the (a-b) F-SS, and (c-d) S-SS after the ammonia immersion and chemical vapor deposition; Figure S3: FE-SEM images of the (a) F-SS and (b) S-SS after the simulated condensation experiments; Figure S4: Equivalent circuits for the EIS results of the (a) SS and DS, (b) BS; Video S1: The coalescence-induced droplet jumping behavior of the F-SS and the S-SS. References [58–60] are cited in Supplementary Materials.

**Author Contributions:** Formal analysis, Z.C., X.C. and Y.S.; investigation, Z.C., X.C. and Y.S.; data curation, Z.C., X.C. and Y.S.; writing—original draft preparation, Z.C., X.C. and Y.S.; Conceptualization, G.W. and P.W.; methodology, G.W. and P.W.; supervision, G.W. and P.W.; funding acquisition, P.W. All authors have read and agreed to the published version of the manuscript.

**Funding:** This research was funded by the National Natural Science Foundation of China (grant number 52261045), the Postdoctoral Innovation Project of Shandong Province (grant number SDBX2021006), and the Natural Science Foundation of Nantong City (grant number JC12022099).

**Data Availability Statement:** The data that support the findings of this study are available from the corresponding author upon reasonable request.

**Conflicts of Interest:** The authors declare no conflict of interest.

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
