# Peer review of "Effect of Microstructure on Coalescence-Induced Droplet Jumping Behavior of a Superhydrophobic Surface and Its Application for Marine Atmospheric Corrosion Protection"

_metals, doi:10.3390/met13081413_

Round 1

Reviewer 1 Report

1. (Micromorphology) Figure 2 shows significant difference in microstructure between F-SS and S-SS specimens. Why the flower-like structure transferred to sheet-like one with prolong soaking treatment? Mechanism?

2.    (Line 243-273) contained too many wordy sentences. The authors should pay more attention on the fitting results. For example the variation of Rf and Rct for different specimens. In addition, please provide the +/- scatter of values.

3.    (Line 262)” a lot of information has been blocked or missing in the Table 1. In addition, parameters derived from electrochemical corrosion data almost never have more than 3 significant digs, often only 2.

4.   Surface morphology observation after condensation and after EIS test is required.

Reviewer 2 Report

The article is devoted to the creation of superhydrophobic coatings on the surface of copper substrates, the authors discuss the coalescence of water droplets depending on the morphological features of the coating. The article is interesting, but contains a number of inaccuracies. This draft is also a preprint (Chen, Zhengshen and Chen, Xiaotong and Wang, Guoqing and Sun, Yihan and Wang, Peng, Effect of Microstructure on Coalescence-Induced Droplet Jumping Behavior of a Superhydrophobic Surface and its Application for Marine Atmospheric Corrosion Protection. Available at SSRN: https://ssrn.com/abstract=4458316 or http://dx.doi.org/10.2139/ssrn.4458316), please explain!

1)      It is not clear from the abstract what the authors mean by the term "droplet jump". Also, the abstract text contains a lot of abbreviations and is very similar to conclusions.

2)      In the introduction, it is necessary to describe in more detail the work in this area: in the scientific literature there are articles devoted to the etching of copper and the preparation of superhydrophobic coatings on these surfaces. For example, the article doi:10.1016/j.corsci.2016.04.015 is very similar to yours, explain what is the difference and what is the novelty of your work?

3)      Presented in fig. 6 results are not entirely clear to the authors. The droplets coalesce, but no “jumps” are visible, a video needs to be added and this effect discussed in more detail. How was "wetting coverage" calculated? Will there be spontaneous removal of water microdroplets if the surface of the sample is tilted?

4)      The conclusions do not match the title, the main idea is "effect of microstructure on coalescence-induced droplet jumping behavior of a superhydrophobic surface", but this is not traceable in the text (only one small section).

5)      Clause 3.5. In Fig. 10, the mirror surface is not visible. The authors claim that condensation leads to degradation of properties (Fig. 9a), explain why? (There is desorption of the hydrophobic agent or a metastable state with a spontaneous transition to homogeneous wetting.) Did you dry the samples before testing?

6)      Table 1 is not readable.

7)      Fig. 6 is not informative, delete it, the data can be given in the text.

Reviewer 3 Report

Although the proposed manuscript is interesting, there are enough weaknesses that need to be improved. This based on the following:

·        The authors should better review the title is very long

·        Line 13-27: the abstract should be reviewed again because it is very general and the objective is not clear. The abstract seems to be the background. What does the acronym EIS stand for? = electrochemical impedance spectroscopy

·        The scope of the study is not well defined, the authors could better express it in the abstract

·        Authors in keywords must select WORDS and not phrases:  marine atmosphere corrosion protection; coalescence-induced droplet jumping behavior; micro structure; solid-liquid contact area.

·        The introduction is very poor, it must be enriched and the authors must integrate more up-to-date literature.

·        Line 51-59The target needs to be better restructured.

·        Line 80-89: Section 2.3 Characterization, The paragraph should separate each analysis technique, FESEM, DRX and XPS

·        What type of detectors was used in the FESEM analysis? secondary or backscattered electrons.

·        The authors must justify why evaluate these materials in 3.5% NaCl? Also, because that frequency range was used for EIS, what is the repeatability of the tests?

·        The authors must correct the title of section 3.1 since there is no micromorphology, the correct thing is morphology, when analyzed by SEM.

·        Figure 2 its legend should be: Figure 2. FE-SEM morphology of .....

·        The images in figure 2 were taken with secondary or backscattered electrodes.

·        In the XPS results, the peak binding energies values must be reported in a results table, likewise the found compound and FWHM fit param must be integrated.

·        In figure 8, the Nyquist diagrams, the axes should be square to avoid flattening the semicircles and the behavior looks much better. Likewise, in the legend of figure 8, it should be indicated which graphs are Nyquist diagrams and which are Bode diagrams.

·        The values of the Y table, they must be corrected, they look very crowded

·        The results only present description of the images and graphics. The results should be discussed with articles from the literature and see what the research is contributing.

·        The conclusions are very poorly stated, it is not possible that the authors do not think of the reader. Putting conclusions in a paragraph is not correct, the conclusions must be specific and not extensive.

·        The authors present 56 references.  11 articles belong to the authors, this is auto plagiarism (19%), only 10% of the total references (5.6 articles) are recommended. Check punctuation, also abbreviations in the references.

Round 2

Reviewer 1 Report

Comments and suggestions have been positively addressed. The manuscript can be accepted in its current revised form.

Reviewer 2 Report

The authors did a good job on the text of the article. I agree with the content of the article. There are some minor remarks left:

1) Earlier I wrote: “Fig. 6 is not informative, delete it, the data can be given in the text.". I made a mistake in the drawing number, I meant Fig.5. Why bring a photo of a drop with corners? If you want to keep this picture, then transfer it to additional materials.

2) Articles previously published by you (Liu XH, Wang P, Zhang D, Chen XT. Atmospheric Corrosion Protection Performance and Mechanism of Superhydrophobic Surface Based on Coalescence-Induced Droplet Self-Jumping Behavior. ACS Appl. Mater. Interfaces 2021;13:25438 -50) are almost completely identical in structure, differing only in the substrate. I think it is necessary in the introduction to talk about your previous research in this area.

Reviewer 3 Report

·        After having reviewed the corrections of the authors, this article can be considered for publication if they attend to the last details.

·        Line 69-81: In the last paragraph of the introduction section, authors must indicate a general objective and not an explanation in a paragraph of more than 12 lines.

·        In the XPS results (Table 1)   the found compound and FWHM fit param must be integrated.

·        The authors must justify why evaluate these materials in 3.5% NaCl? Also, because that frequency range was used for EIS, what is the repeatability of the tests?

·        Line 364-370: This paragraph should be deleted, it is not a conclusion

·        The authors present 57 references.  The authors corrected self-plagiarism
